# Tsinghua Scientific Satellite Precise Orbit Determination Using Onboard GNSS Observations with Antenna Center Modeling

**Kai Shao [1], Chunbo Wei [1], Defeng Gu [2,\*], Zhaokui Wang [3], Kai Wang [1], Yingkai Cai [3] and Dachen Peng [1]**

[1] School of Physics and Astronomy, Sun Yat-sen University (Zhuhai Campus), Zhuhai 519082, China; shaok3@mail.sysu.edu.cn (K.S.); weichb3@mail2.sysu.edu.cn (C.W.); wangk273@mail2.sysu.edu.cn (K.W.); pengdch@mail2.sysu.edu.cn (D.P.)

[2] School of Artificial Intelligence, Sun Yat-sen University (Zhuhai Campus), Zhuhai 519082, China

[3] School of Aerospace Engineering, Tsinghua University, Beijing 100084, China; wangzk@tsinghua.edu.cn (Z.W.); cyk19@mails.tsinghua.edu.cn (Y.C.)

\* Correspondence: gudefeng@mail.sysu.edu.cn; Tel.: +86-137-8712-9336

**Abstract:** The Tsinghua scientific satellite is a Chinese spherical micro satellite for Earth gravity and atmospheric scientific measurements. The accurate orbits of this satellite are the prerequisites to satisfy the mission objectives. A commercial off-the-shelf dual-frequency GNSS receiver is equipped on the satellite for precise orbit determination (POD). The in-flight performances of the receiver are assessed. Regular long-duration gaps up to 50 min are observed in GNSS data, and the typical data availability is about 60–70% each day. The RMS of code noises is 0.24 m and 0.30 m for C1 and P2 codes, respectively. The RMS of fitting residuals of the carrier phase geometry-free L1–L2 combination is 2.4 mm. The GNSS receiver antenna center offsets (ACOs) and antenna center variations (ACVs) maps are estimated using in-flight data for both dual-frequency and single-frequency POD. Significant improvements in POD performances are obtained when the measurement models are updated by using the ACO and ACV maps' corrections. With the updated measurement model, the RMS of the orbit overlap differences is 1.23 cm in three dimensions for dual-frequency POD, which is reduced by 27%. Meanwhile, two different empirical acceleration types are employed and compared for dual-frequency POD, and the results show that consistency on the 5 cm level is demonstrated for orbit solutions obtained with the updated measurement model. After correcting the ACO and ACV maps, the precision of single-frequency orbit solutions is better than 10 cm, which is improved by 32%. The results indicate that the antenna center modeling can significantly improve the consistency of Tsinghua scientific satellite precise orbits, which will be conducive to the realization of the mission objectives.

**Keywords:** Tsinghua scientific satellite; precise orbit determination; spaceborne GNSS; antenna center modeling; precision assessment

## 1. Introduction

The Tsinghua scientific satellite, which aims to achieve technology verification of upper atmospheric density and Earth gravity field detection, was developed by Tsinghua University. It was successfully launched on 6 August 2020 into low Earth orbit (LEO) with an altitude of about 500 km. The micro satellite is designed in a spherical configuration with a diameter of 626 mm and mass of approximately 22 kg. High-precision orbit data can be used to recover atmospheric density and gravity fields with a high temporal resolution and accuracy [1–3]. To satisfy the mission objectives, the initially specified orbit accuracy is at the cm level [4], and a commercial off-the-shelf dual-frequency GNSS receiver is equipped on the satellite for precise orbit determination (POD).

The reduced-dynamic method, which requires a pseudo-stochastic parameterization to compensate potential deficits in the employed force models, has been widely applied for generating precise orbital products of LEO satellites [5,6]. Many efforts have been made in reduced-dynamic POD for large-size satellites, which usually were equipped with high-quality dual-frequency GNSS receivers. The dual-frequency GNSS observations have allowed for the POD to reach up to 3–5 cm for many LEO satellites, such as CHAMP [7], SWARM [8], GRACE [9], and GRACE-FO [10] satellites. With the development of micro satellite industry, more attentions have been paid to the POD of micro satellite equipped with low-cost GNSS receivers. Due to the availability of GNSS observations is restricted by the quality of storage and downlink of some micro satellite missions, the POD accuracy is at the cm level to dm level when using dual-frequency GNSS observations [11,12]. Meanwhile, GNSS single-frequency orbit determination for LEO micro satellites with low-cost GNSS receivers has also been widely researched [13]. The single-frequency GNSS data are considered when only single-frequency data are available or the observation qualities of dual-frequency data are quite different [14]. In addition, single-frequency POD can be used as an alternative to dual-frequency POD. Depending on the code data qualities, the precision of single-frequency POD based on the group and phase ionospheric correction (GRAPHIC) combinations is at the dm level for the small and micro satellite missions, such as APOD-A [15] and Loujia-1A [16] satellites. For the Tsinghua scientific satellite POD, the performance of the commercial off-the-shelf dual-frequency GNSS receiver was tested on ground by using GNSS signal simulator. The results show that the precision of the satellite orbit could be better than 5 cm. Due to the differences between the ground test and test on the orbit environment, it is necessary to study the in-flight tracking as well as dual-frequency and single-frequency orbit determination performances of the receiver, which are important for the satellite's Earth gravity and atmospheric measurements.

The receiver antenna centers of carrier phase and code observations play a crucial role in GNSS data processing [17,18]. For dual-frequency orbit determination, high-precision orbit solutions primarily rely on the GNSS carrier phase observations. It is essential to apply receiver antenna phase center offset (PCO) and phase center variations (PCVs) in GNSS-based dual-frequency orbit determination, where the code center offset and variation of the receiver antenna are less significant due to the much lower weights of code observations. The methods of antenna center modeling for GNSS observations have been widely developed in dual-frequency orbit determination, the receiver antenna PCOs can be directly estimated, and the PCVs maps are usually obtained by the residual approach [19,20]. The impacts of GNSS antenna center modeling on LEO satellite dual-frequency POD have been studied, such as GRACE [21], GOCE [22], HY-2A [23], and ICESat-2 [24] satellites. For single-frequency orbit determination, since the GRAPHIC combinations are used, the same as receiver antenna PCO and PCVs, the equivalent receiver antenna GRAPHIC residual offset (GRO) and GRAPHIC residuals variations (GRVs) are modeled and applied in single-frequency POD for several LEO satellites [25]. The application of receiver antenna GRO and GRVs, which can mainly eliminate the systematic errors of GRAPHIC observations, could further enhance single-frequency orbit solutions. Since both dual-frequency and single-frequency POD for the Tsinghua scientific satellite are considered, we use the terms antenna center offset (ACO) and antenna center variations (ACVs) in this work. For improving both dual-frequency and single-frequency orbit solutions of the Tsinghua scientific satellite, the GNSS receiver ACOs and ACV corrections are estimated using in-flight data, and the improvements on the Tsinghua scientific satellite POD are investigated.

In this paper, we focus on the study of the Tsinghua scientific satellite precise orbit determination using onboard GNSS observations with antenna center modeling. Following the general information of the satellite and GNSS receiver, the tracking and measurement performance of the GNSS receiver are assessed. Subsequently, both the dual-frequency and single-frequency reduced-dynamic POD strategies with antenna center modeling are introduced, and the results of ACOs and ACVs maps and their impacts on dual-

frequency and single-frequency POD are presented, respectively. Then, we discussed the POD results in terms of POD post-fit residuals, estimated empirical accelerations and scaling parameters, and orbit consistency. Finally, the conclusions of this study are presented.

## 2. Spaceborne GNSS Data Assessment

### 2.1. Satellite and Receiver Feature

An artist's impression of the Tsinghua scientific satellite including satellite reference frame (SRF) is shown in Figure 1. The SRF is defined as follows. The origin is the geometric center of the satellite, which is the center of the sphere. The $Z_{SRF}$ axis is positive in the direction of the satellite radial direction, and the $X_{SRF}$ axis is perpendicular to the $Z_{SRF}$ axis in the orbital plane and is along the satellite velocity direction, while the $Y_{SRF}$ axis completes the right-handed orthogonal coordinate system. An antenna-fixed coordinate system (ARF) is also defined for antenna center modeling. The origin is the GNSS receiver antenna reference point (ARP), and the $X_{ARF}$ is along the $X_{SRF}$. While the $Z_{ARF}$ is along the antenna zenith direction, which rotates 180° with the $Z_{SRF}$ axis.

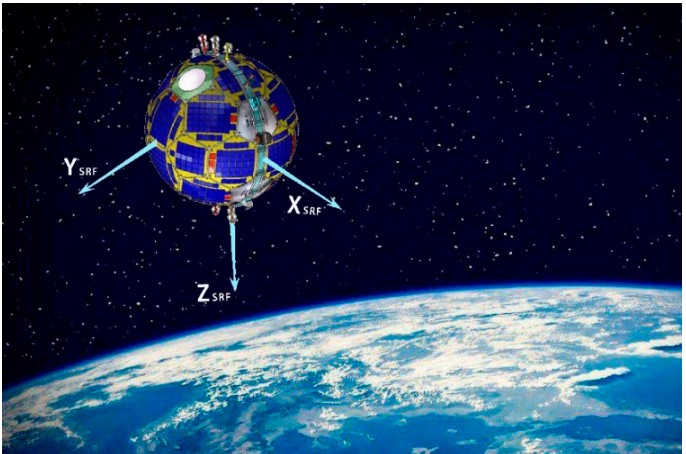

**Figure 1.** Artist's impression of the Tsinghua scientific satellite.

Figure 2 shows the commercial off-the-shelf dual-frequency GNSS receiver with its matched microstrip antenna. The GNSS receiver antenna is mounted on the $Y_{SRF}$ and $Z_{SRF}$ axis plate with a tilted azimuth about 31.72° along the $Z_{SRF}$ axis. The receiver can concurrently track the GPS signal at the L1 and L2 frequencies and provide both code and carrier phase observations. The approximate coordinates of the receiver ARP and the center of mass (CoM) in the SRF system are given in Table 1.

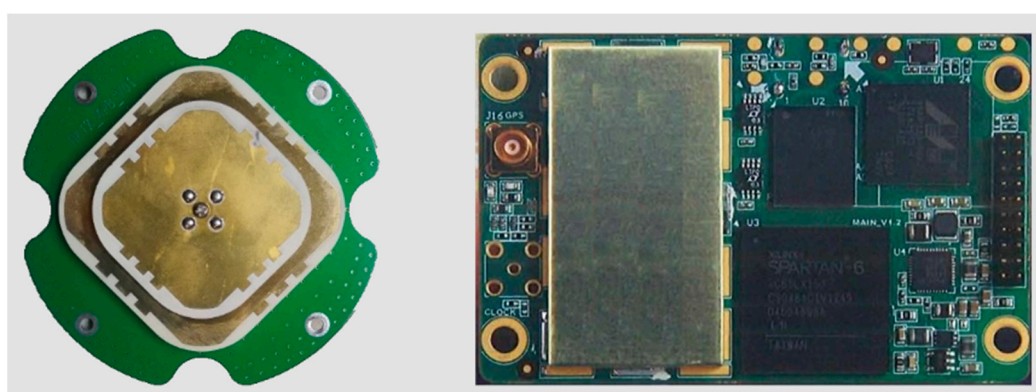

**Figure 2.** The commercial off-the-shelf GNSS receiver (**right**) and microstrip antenna (**left**) for the Tsinghua scientific satellite.

**Table 1.** Approximate coordinates of the satellite CoM and GNSS receiver ARP.

| ID | $X_{SRF}$ (m) | $Y_{SRF}$ (m) | $Z_{SRF}$ (m) |
|----|------|------|------|
| CoM | 0.0037 | −0.0628 | 0.0094 |
| ARP | 0.0000 | 0.1268 | −0.2080 |

The sun sensor and magnetometer are considered as attitude sensors for satellite attitude control. A three-axis stable attitude determination and control system (ADCS) is employed in the Tsinghua scientific satellite. The accuracy of the ADCS for the attitude control is approximately 3°. We used the nominal three-axis stable attitude instead of the actual attitude information in the following analysis.

### 2.2. Tracking Ability of GNSS Satellites

GPS observation data from 11 October to 21 October (day of year (DOY) 285–306), 2020 were used to analyze the GPS satellite tracking ability of the receiver. The decoded RINEX files (RINEX 2.10) including GPS code measurements (C1 and P2) and carrier phase measurements (L1 and L2) with 4 s intervals were obtained from Tsinghua University.

Figure 3 shows the sky coverage of the tracked GPS satellites for DOY 285 in 2020. The hard-coded elevation is limited to 5° relative to the local horizon in the local orbital frame system. In the left of the flight direction, the data are lost at low elevations due to the tilted installation of receiver antenna, which is similar to International Space Station [26] and Tiangong-2 [27].

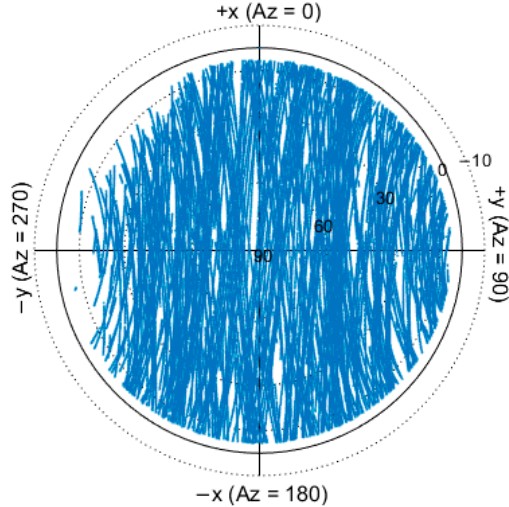

**Figure 3.** Sky coverage of the observed GPS satellites in the SRF system.

Figure 4 shows the GPS tracking arcs along with the elevations for DOY 285 in 2020. It is found that the original observation data have multiple interruptions, which may be due to the fact that the limited system resources restrict the quantity of storage and downlink observations of GPS measurements. During the test period, regular long-duration gaps up to 50 min were observed, and the typical data availability was about 60–70% each day. The interruption of more than 30 min accounted for about 30% of the total number of interruptions. The average tracking time per day was about 15.67 h.

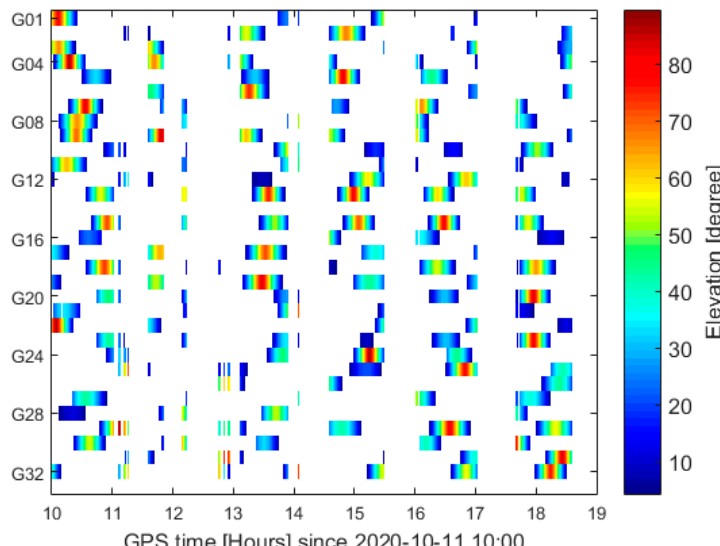

**Figure 4.** Duration and elevation of continuous tracking arcs for GPS satellites with interruptions.

For GPS L1 and L2 observations, the distribution of the numbers of visible satellites in the epoch is shown in Figure 5. The receiver can track up to 13 GPS satellite at the same time. The number of visible satellites on L1 frequency is bigger than that on the L2 frequency, and the average numbers of visible satellites are 9.5 and 8.0, respectively.

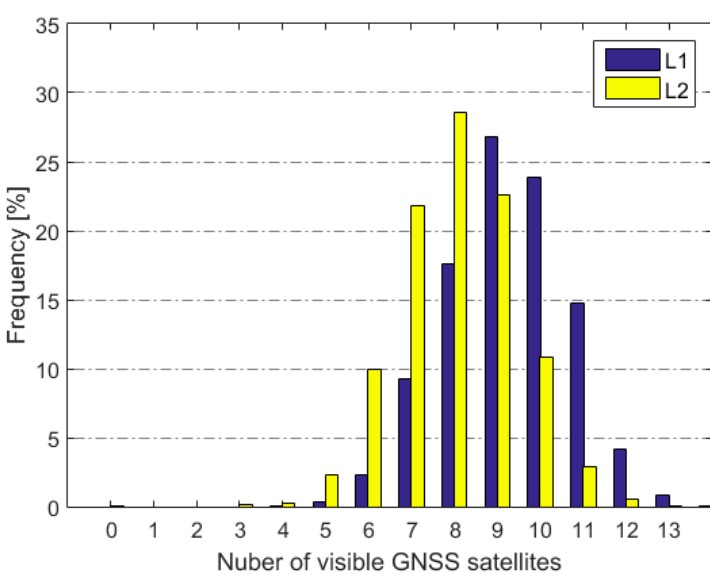

**Figure 5.** Statistics of visible GPS satellites on L1 and L2 frequency.

The sky plots of the carrier-to-noise density ratio ($C/N_0$) for L1 and L2 frequency in the SRF system are shown in Figure 6. The mean values of $C/N_0$ were obtained in each bin with the resolution of 3° × 3°. As the GNSS receiver antenna is tilted azimuth about 31.72° and the elevation-cutoff threshold of the receiver is defined at 5°, the $C/N_0$ map exhibits a rather patchy structure and shows pronounced areas in which the signal strength is affected by shadowing of satellite body. The mean $C/N_0$ values are 43 and 31 for the L1 and L2 frequency, respectively. The $C/N_0$ for L2 frequency is limited to 20 and much lower than that for the L1 frequency.

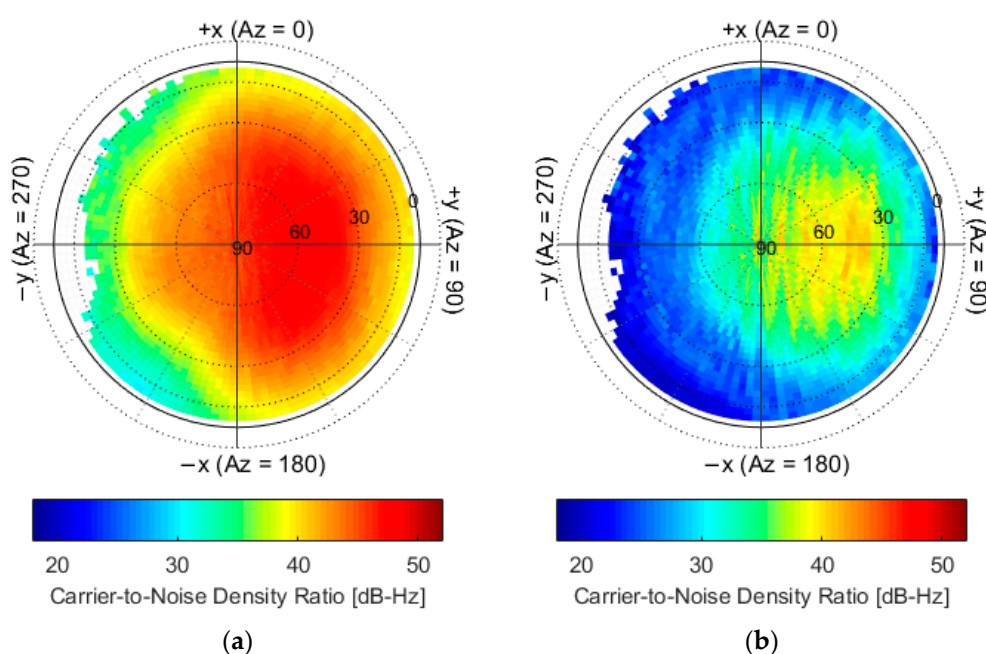

(**a**)            (**b**)

**Figure 6.** Carrier-to-noise density ratio for GPS L1 frequency (**a**) and L2 frequency (**b**) in the SRF system.

### 2.3. Quality of GNSS Observations

The qualities of GPS code and carrier phase observations obtained from the receiver were assessed. We used the geometry and ionosphere-free multipath combination (MPC) to analyze the code accuracy. Following the methodology outlined in Gu [15], the multipath system errors of the C1 and P2 codes were obtained and are shown in Figure 7. The mean values of MPC were obtained in each bin with the resolution of 5° × 5°. The multipath errors exhibit a clear characteristic of systematic deviation, and a systematic deviation is found in high elevation bin up to 90°. The multipath errors in the low and high elevation bins for P2 code are much bigger than that for the C1 code.

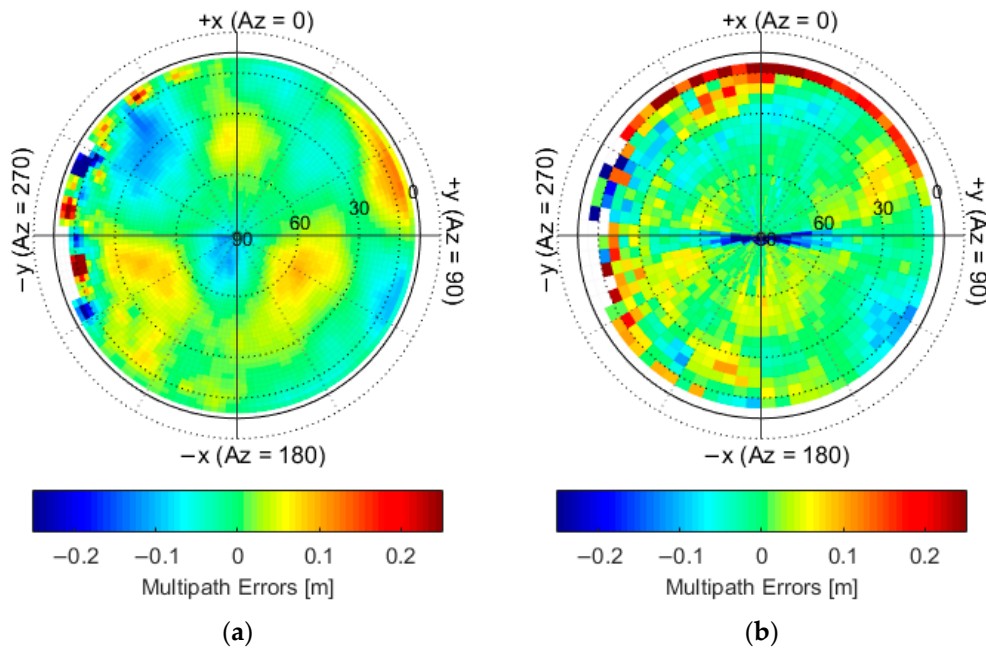

(**a**)            (**b**)

**Figure 7.** Multipath error maps of C1 code (**a**) and P2 code (**b**) given in the SRF system.

The code noises are then obtained by subtracting multipath errors from the MPCs in each azimuth and elevation bin. The RMS of the C1 and P2 code noises as a function of elevation is shown in Figure 8. The P2 code noises are bigger than that of C1 code, and the average RMS of the code noises is 0.24 m and 0.30 m for C1 and P2 codes, respectively. The level of code noises is equivalent to the Chinese Tiangong-2, TH-2 [28] and TianQin-1 [29] satellites, but significantly better than that of the APOD-A satellite [15] for the P2 code.

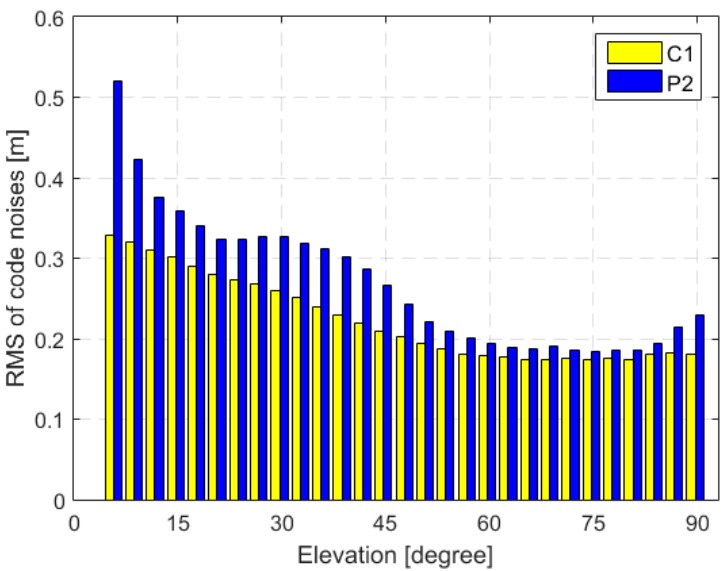

**Figure 8.** Variations in RMS of the C1 and P2 code noises with elevation.

The carrier phase accuracy is assessed using L1–L2 combinations. A fourth-order piecewise polynomial smoothing algorithm over a sliding 60 s interval is used to fit the data. The daily RMS of fitting residuals of the L1–L2 combinations is shown in Figure 9. The average RMS of about 2.4 mm was obtained for the L1–L2 combinations, which is similar to the Tiangong-2 and TH-2 satellite, but smaller than the TianQin-1 satellite.

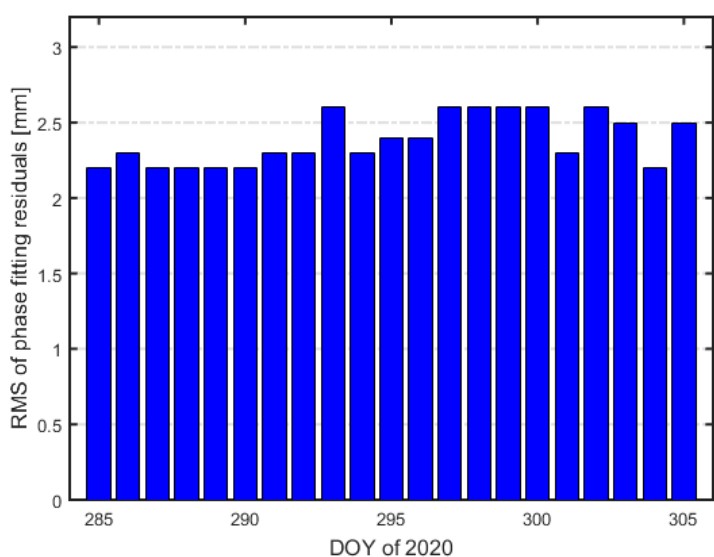

**Figure 9.** RMS of fitting residuals of the L1–L2 combinations.

## 3. Orbit Determination and Antenna Center Modeling

### 3.1. Orbit Determination Strategy

For the Tsinghua scientific satellite POD, the dual-frequency and single-frequency data were used to obtain the dual-frequency orbit solution and single-frequency orbit solution, respectively. Although the commercial off-the-shelf dual-frequency GNSS receiver is equipped on Tsinghua scientific satellite, it is still worthy to analyze the performance of single-frequency orbit solution because the observation quality of the L1 frequency of the receiver is much better than that of the L2 frequency, and the analysis could contribute to a better understanding of micro satellite applications with a commercial off-the-shelf single-frequency GNSS receiver. In this study, the ionosphere-free (IF) combination carrier phase observations ($L_{IF}$) and GRAPHIC combination observations ($\rho_G$) were used for dual-frequency and single-frequency POD, respectively.

$$L_{IF}^{j} = \frac{f_1^2}{f_1^2 - f_2^2} L_1^{j} - \frac{f_2^2}{f_1^2 - f_2^2} L_2^{j} = \rho^{j} + c \cdot (\delta t_r - \delta t^{j}) + \lambda_{IF} N_{IF}^{j} + \varepsilon_{IF} \tag{1}$$

$$\rho_G^{j} = \left( C_1^{j} + L_1^{j} \right)/2 = \rho^{j} + c \left( \delta t_r - \delta t^{j} \right) + \lambda_1 N_1^{j}/2 + \varepsilon_G \tag{2}$$

where subscripts 1 and 2 denote different frequencies, superscript $j$ denotes the $j$-th GPS satellite, $r$ denotes the receiver, $L_i$ is the carrier phase observation, $C_1$ is $C_1$ code observation, $f_i$ is the carrier frequency, $\rho$ denotes the geometric distance from the LEO satellite to the GPS satellite, $c$ is the speed of light, $\delta t_r$ is the receiver clock error, $\delta t^j$ is the GPS satellite clock error, $\lambda$ is the wavelength, $N_1$ is the carrier phase ambiguity of $L_1$, and $N_{IF}$ is the ambiguity of IF carrier phase combination; $\varepsilon$ contains measurement noise, multipath error, and all other unmodeled errors.

The National University of Defense Technology orbit determination toolkit (NUDTTK) software [30], which has proven ability for the LEO satellite POD, was employed in the Tsinghua scientific satellite POD. A summary of the measurement and dynamical models used in the NUDTTK software is presented in Table 2. The reduced-dynamic POD mothed was employed for the satellite POD with a typical process time length of 30 h from 21:00 on the previous day to 03:00 on the next day. The atmospheric drag, solar radiation pressure coefficients, and empirical accelerations as well as initial orbit parameters were estimated by a batch least-squares estimator. The float ambiguities for IF carrier phase were estimated, and the ambiguities were not fixed in the following analysis. Due to different measurement errors of IF carrier phase and GRAPHIC combination observations, the piecewise constant accelerations with constraints ($1.0 \times 10^{-6}$ m/s²) for dual-frequency POD and with constraints ($2.0 \times 10^{-7}$ m/s²) for single-frequency POD were employed. The GPS receiver antenna centers for IF carrier phase combination and GRAPHIC combination were corrected by antenna center modeling, which is introduced next.

**Table 2.** Measurement and dynamical models employed in the NUDTTK software for the Tsinghua scientific satellite POD.

| Model | Description |
| --- | --- |
| GPS measurement model | |
| GPS observations | Carrier phase IF combination for dual-frequency POD, GRAPHIC combination for single-frequency POD; Undifferenced observations with 8 s sampling; 30 h arc length |
| GPS orbit and clocks | CODE final products with 30 s clock sampling [31] |
| GPS antenna corrections | IGS igs14.atx [32] |
| Phase windup | Applied [33] |
| GPS data weighting | With function 2sin(θ) when elevation angle below 30° |
| Receiver antenna corrections | A priori APR and antenna center modeling |
| Gravitational force models | |



| | |
|---|---|
| Earth gravity field | GGM05C 180 × 180 [34] |
| Ocean tides | EOT11a 60 × 60 [35] |
| Solid Earth and pole tides | IERS 2010 [36] |
| Third body gravity | Luni-solar-planetary gravity, DE430 [37] |
| Relativity | IERS 2010 |
| Non-gravitational force models | |
| Spacecraft surface model | Cannon ball model with an area-to-mass ratio of 0.01 m²/kg |
| Atmospheric density model | Jacchia-71 [38] |
| Atmospheric drag coefficients | Estimated every 3 h |
| Solar radiation pressure | Conical Earth shadow model; one coefficient estimated for whole arc |
| Empirical acceleration | Piecewise constant accelerations in radial (R), along-track (T) and cross-track (N) directions: 10 min intervals |

*3.2. Antenna Center Modelling*

For LEO satellite POD, proper modeling of GNSS observations is a prerequisite, and the distances of where GNSS signals enter the LEO satellite receiver antenna relative to the center of mass (COM) should be known as accurately as possible. The differences between the electric center of the antenna and the physically defined point inside the antenna are often divided into two parts. The first part is defined as the ACO vector, which means a frequency-dependent offset from an average antenna center. The second part is defined as ACVs, which depend on the elevation and azimuth.

The differences are commonly denoted as PCO and PCVs in dual-frequency orbit determination, which primarily rely on the GNSS carrier phase observations. For many LEO satellites, the priori PCO vectors and PCVs pattern can be obtained from ground calibrations before launching, which are often significantly different from in-flight calibration due to the ground calibration error, fuel consumption, and in-flight environment variations. In high-precision dual-frequency orbit determination application, in-flight calibration and compensation for antenna PCO vectors and PCVs are necessary. In general, corresponding ACO vectors and ACVs patterns are missing for code observations with ground or in-flight calibrations, since their impacts on dual-frequency orbit determination are much less significant [19,20]. However, similar to the receiver antenna PCO and PCVs in dual-frequency orbit determination, the receiver antenna center corrections of GRAPHIC combinations consist of the GRO vectors, and GRVs are existent in single-frequency orbit determination when the GRAPHIC linear combinations are used [25].

For the Tsinghua scientific satellite, because only the receiver ARP is well defined, and no ACO vectors and ACVs patterns are obtained, it is important to estimate the ACO vectors and ACVs patterns using in-flight data. Therefore, the ACO vectors and ACVs patterns are estimated for both dual-frequency and single-frequency orbit determination.

The ACO vectors were estimated first. Given that the Tsinghua scientific satellite was operated in the three-axis stable attitude control mode, we only estimated the z-component parameter of ACO according to the methodology outlined in Gu [18]. It should be noted that, due to the correlations between piecewise constant accelerations in the radial and the z-component parameter of ACO, when we estimated the z-component parameter of ACO, the piecewise constant accelerations in the radial component were not estimated simultaneously. Figure 10 shows the daily receiver ACO estimation results in z-component for dual-frequency IF carrier phase combinations and single-frequency GRAPHIC combinations. The mean values of ACO for dual-frequency and single-frequency combinations are 2.04 cm and 1.71 cm, respectively. The results are stable daily with a standard deviation of 0.19 cm for the dual-frequency IF carrier phase combinations. However, the standard deviation of ACO results is 0.72 cm for the single-frequency GRAPHIC combinations. The final ACO corrections are the mean value of daily antenna ACO error estimation results. The impacts of ACO corrections on LEO satellite POD are mainly in the R direction, which are almost constant offsets for the orbit solutions.

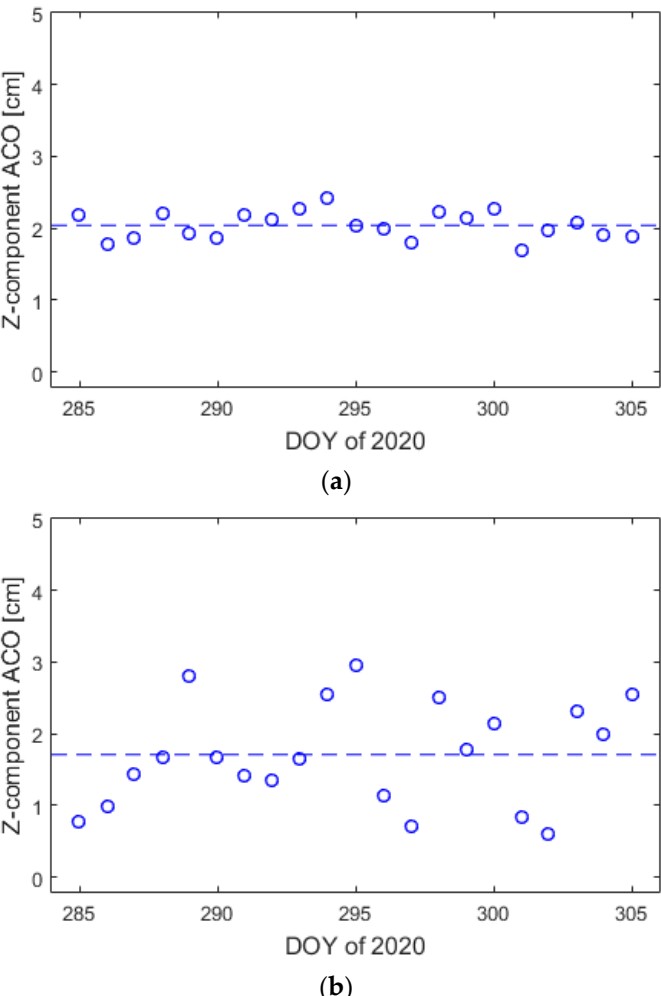

**Figure 10.** Daily ACO estimation results in *z*-direction for dual-frequency IF carrier phase combinations (**a**) and single-frequency GRAPHIC combinations (**b**). The blue dashed lines denote the average values, and the blue circles denote the daily z-component ACO values.

The receiver ACVs were generated based on the observation residuals from POD in an in-flight calibration. The ACVs for dual-frequency IF carrier phase combinations and single-frequency GRAPHIC combinations both were mapped in the azimuth/elevation bins of 5° × 5°. This initial map was introduced in a first iteration step (N1) of the reduced-dynamic POD. Then, the dual-frequency IF carrier phase observation and single-frequency GRAPHIC observation residuals were corrected for the initial ACVs maps, respectively. The mean values of the observation residuals of the new orbit solutions were used to improve the initial ACV map.

Figure 11 shows the three-dimensional (3D) RMS of orbit differences between the solution with ACV maps (Ni) and no ACV maps (N0). The impacts of ACV maps in dual-frequency and single-frequency POD were pronounced. After four iterations, the impacts were almost the same.

The final ACVs maps were obtained after four iterations, which are shown in Figure 12. Both maps have the characters of systematic deviations. For the ACVs map of dual-frequency IF carrier phase observations, the scale was limited to −22 to 22 mm but extreme values were −35.0 to 56.2 at low elevations, respectively. For ACVs map of single-frequency GRAPHIC observations, the scale was limited to −102 to 102 mm but extreme values were −513.6 to 299.9 at low elevations, respectively.

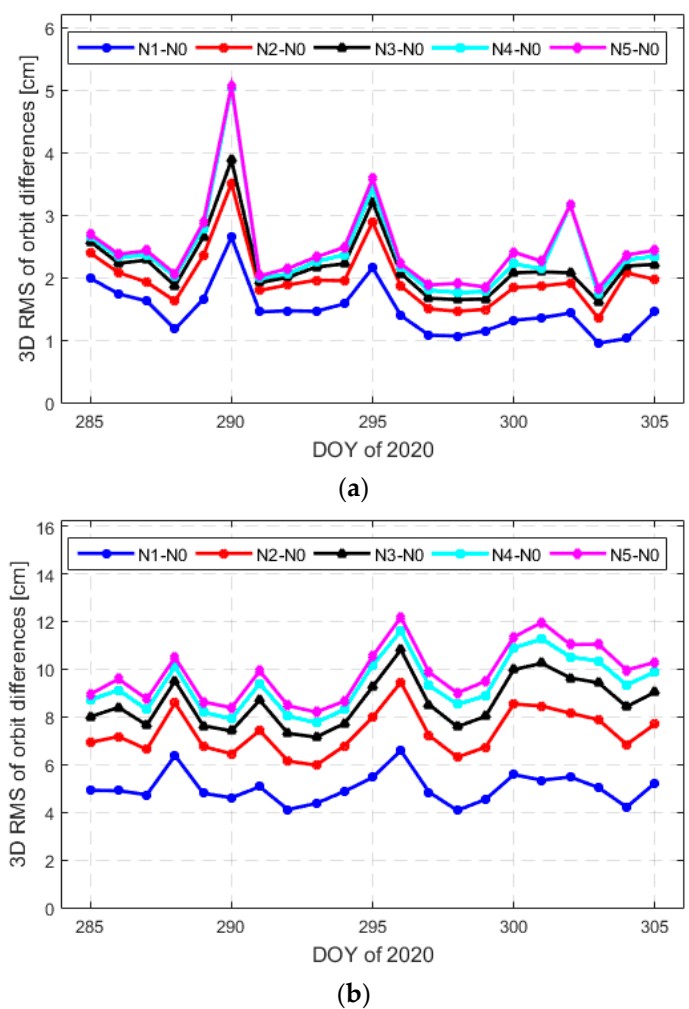

**Figure 11.** Impacts of ACV maps in reduced-dynamic POD for dual-frequency IF carrier phase combination (**a**) and single-frequency GRAPHIC combination (**b**).

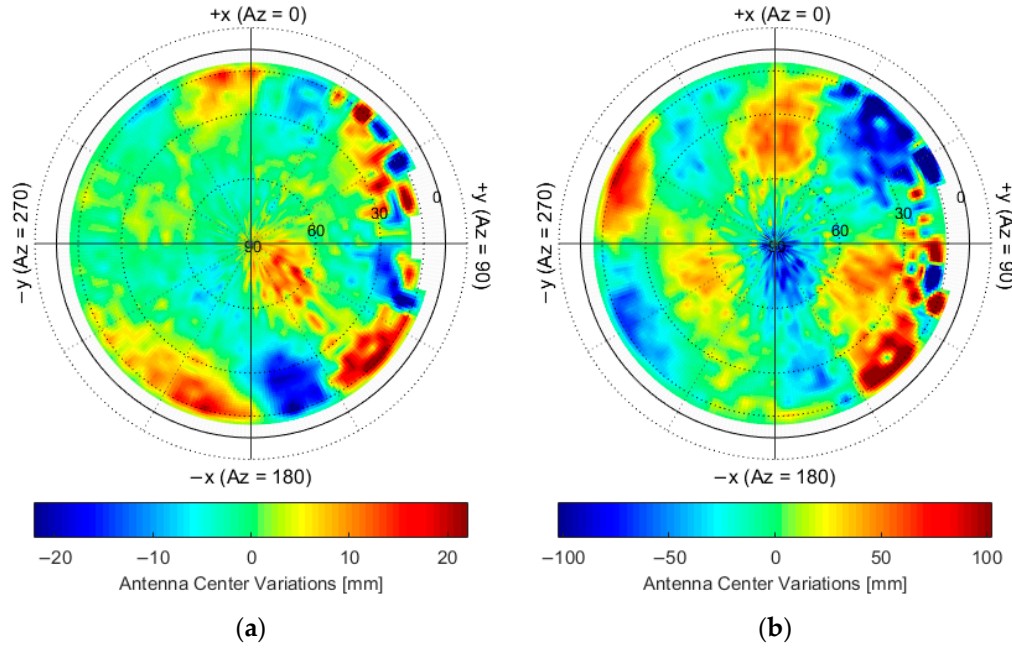

**Figure 12.** Final estimated ACVs maps from dual-frequency POD (**a**) and single-frequency POD (**b**).

## 4. POD Results

Three difference sets of POD solutions were generated, which are denoted as A priori, +ACO, and Updated, corresponding to the orbit solutions using the receiver ARP without ACO and ACVs maps corrections, with ACO corrections and without ACVs maps corrections, and with ACO and ACVs map corrections, respectively. Both the dual-frequency and single-frequency orbit solutions were obtained and compared without and with ACO and ACV map corrections.

### 4.1. Dual-Frequency POD Results

To evaluate the benefits of antenna center modeling on dual-frequency POD, the results of POD post-fit residuals for dual-frequency IF carrier phase observations, estimated empirical accelerations, and scaling parameters were first analyzed. The POD post-fit residuals can reflect the noise level of observations and the potential deficits in the dynamical models or observation models. Empirical acceleration is a key parameter in the process of reduced-dynamic POD. The magnitude can reflect the potential modeling errors in the dynamical models and observation models and reflect the accuracy and stability of parameter estimation. Additionally, the non-gravitational force models include scaling parameters, i.e., CD for atmospheric drag and CR for solar radiation pressure, which can also reflect the accuracy and stability of parameter estimation in the Tsinghua scientific satellite POD.

Figure 13 shows the RMS of POD post-fit residuals for the dual-frequency IF carrier phase observations as a function of elevation. Without ACO and ACVs map corrections, the RMS value was about 1.6 cm at low elevations, which reduced to 0.8 cm at high elevations. The average RMS value of POD post-fit residuals for the dual-frequency IF carrier phase observations reduced from 1.2 cm to 1.1 cm with the updated ACO and ACVs map corrections.

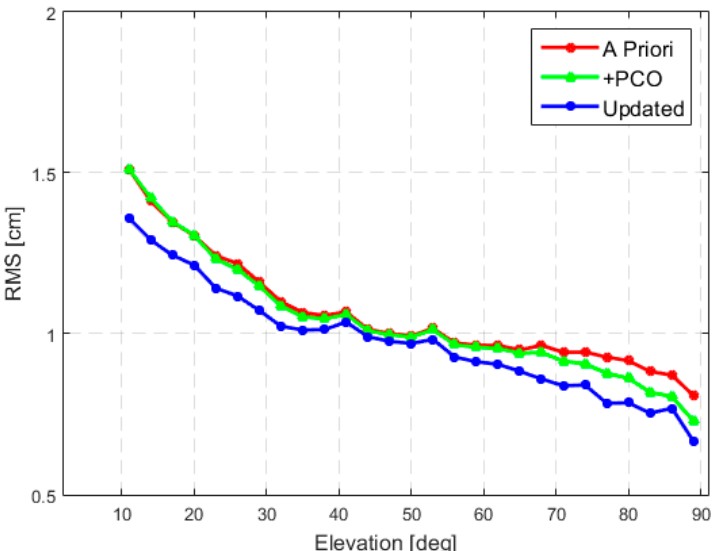

**Figure 13.** RMS of POD post-fit residuals for the dual-frequency IF carrier phase observations as a function of elevation.

The statistical results of estimated empirical accelerations and scaling parameters are listed in Table 3. The use of ACO mainly reduces the empirical accelerations in R direction. The ACVs map can reduce the empirical accelerations in all directions. After using ACO and ACVs map corrections, the RMS values of empirical accelerations were reduced by 56%, 26%, and 7% in the R, T, and N directions, respectively.

Meanwhile, the estimated scaling parameters for atmospheric drag are shown in Figure 14. The atmospheric drag coefficients exhibit almost same mean value but reduced the

standard deviation (STD) value after using ACO and ACVs map corrections. For solar radiation pressure coefficients, the mean and STD values were increased and reduced, respectively. The final average of CR coefficients was close to 1. The use of the updated ACO and ACVs map could improve the stability of estimated parameters from the reduced-dynamic POD.

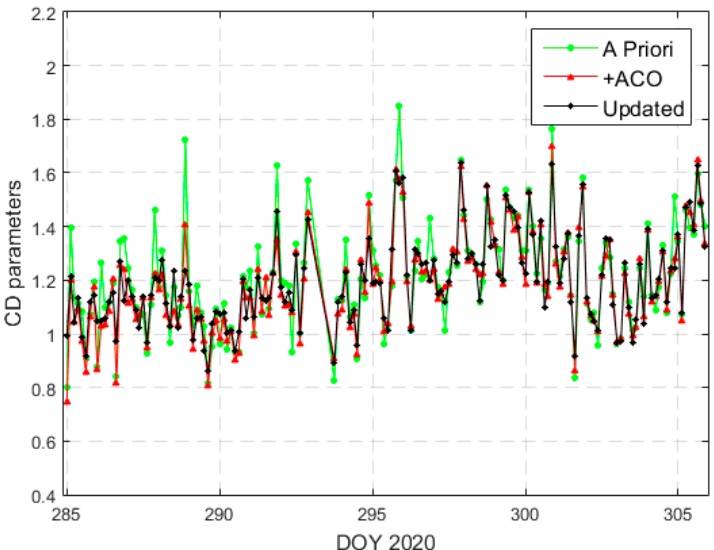

**Figure 14.** Daily estimated CD parameters of atmospheric drag from reduced-dynamic POD.

**Table 3.** Statics of the empirical accelerations and estimated scaling parameters from the reduced-dynamic POD.

| ID | RMS of Empirical Accelerations (nm/s$^2$) | | | CD Mean and STD | CR Mean and STD |
|---|---|---|---|---|---|
| | **R** | **T** | **N** | | |
| A priori | 25.98 | 15.95 | 26.34 | 1.21 ± 0.20 | 0.87 ± 0.06 |
| +ACO | 13.71 | 14.00 | 26.36 | 1.19 ± 0.18 | 1.11 ± 0.04 |
| Updated | 11.49 | 11.73 | 24.41 | 1.20 ± 0.17 | 1.04 ± 0.04 |

For dual-frequency POD, the orbit precisions were assessed by orbit overlapping comparison and orbit comparison between the different orbit solutions obtained by using two different types of empirical accelerations. The 30 h orbit arcs allow for an overlap of 6 h between two consecutive arcs, which is an essential approach to assess the internal orbit consistency. Another empirical acceleration type of 1 cycle-per-revolution accelerations per orbital revolution was also used to obtain dual-frequency POD. The orbits obtained by using two different empirical acceleration types were independent, the differences of which can reflect the orbit precisions.

Table 4 lists the statistical results of orbit overlapping comparison and orbit comparison between different orbit solutions. The daily 3D RMS values of the orbit differences between the different orbit solutions are shown in Figure 15. The RMS of orbit overlapping differences was significantly reduced after applying ACO and ACVs map corrections. The orbit precision improvement was approximately 27%. The result of orbit comparison between different orbit solutions also shows the benefits of the antenna center modeling. The 3D RMS of orbit differences was 4.42 cm when using ACO and ACVs map corrections.

**Table 4.** Statics of the overlapping orbits and differences between different orbit solutions.

| ID | Overlap Orbit Differences RMS (cm) | | | | Orbit Solution Differences RMS (cm) | | | |
|---|---|---|---|---|---|---|---|---|
| | R | T | N | 3D | R | T | N | 3D |
| A priori | 0.89 | 1.29 | 0.59 | 1.68 | 4.42 | 4.58 | 1.98 | 6.69 |
| +ACO | 0.72 | 1.11 | 0.63 | 1.47 | 3.51 | 3.69 | 1.67 | 5.38 |
| Updated | 0.61 | 0.86 | 0.62 | 1.23 | 2.82 | 2.98 | 1.55 | 4.42 |

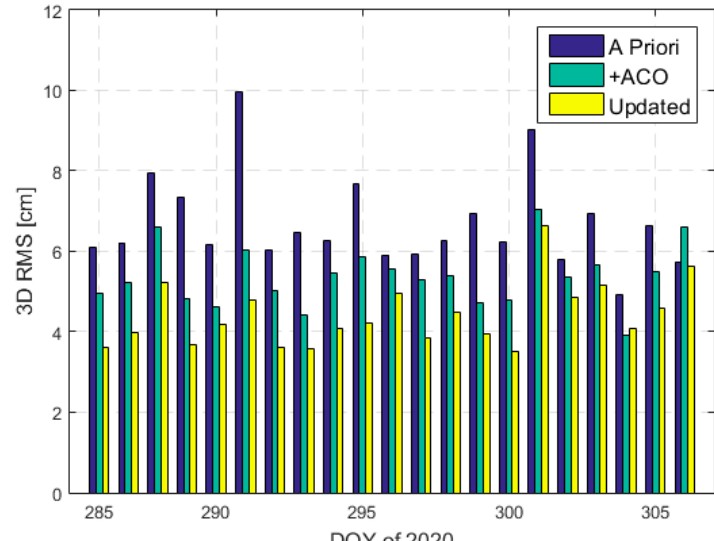

**Figure 15.** Orbit comparison between different orbit solutions obtained by using two different types of empirical accelerations.

### 4.2. Single-Frequency POD Results

Due to large code noises, the POD post-fit residuals for single-frequency GRAPHIC combination observations were much bigger than that for dual-frequency IF carrier phase observations. The RMS of POD post-fit residuals for single-frequency GRAPHIC combination observations reached up to 10.8 cm, which were slightly reduced by 1% and 2% after using only ACO correction and ACO and ACVs map corrections for single-frequency POD, respectively.

The statistical results of estimated empirical accelerations and scaling parameters from the single-frequency POD are listed in Table 5. It also shows that the antenna center modeling for single-frequency POD can reduce the empirical accelerations and improve the stability of estimated parameters. Different to dual-frequency POD, it can be found that the improvements mainly come from ACVs map corrections in single-frequency POD.

**Table 5.** Statistics of the empirical accelerations, estimated scaling parameters from single-frequency POD, and the difference between dual-frequency and single-frequency orbit solutions.

| ID | Emp. Acc. (nm/s²) | | | CD Mean and STD | CR Mean and STD | Orbit Difference RMS (cm) | | | |
|---|---|---|---|---|---|---|---|---|---|
| | R | T | N | | | R | T | N | 3D |
| A priori | 7.60 | 10.48 | 17.47 | 1.24 ± 0.19 | 1.04 ± 0.05 | 4.43 | 12.52 | 3.93 | 13.88 |
| +ACO | 7.49 | 10.47 | 17.32 | 1.23 ± 0.19 | 1.05 ± 0.05 | 4.38 | 12.40 | 3.84 | 13.73 |
| Updated | 5.03 | 7.35 | 13.73 | 1.21 ± 0.16 | 1.08 ± 0.03 | 2.75 | 8.60 | 2.62 | 9.42 |

For validating the precision of single-frequency orbit solutions, the orbit differences with respect to the dual-frequency orbit solutions obtained with ACO and ACVs map

corrections were employed. The statistical results are also listed in Table 5. The daily 3D RMS values of the orbit differences are shown in Figure 16. The ACO corrections have a slight impact on single-frequency POD, but the improvement is remarkable by using ACVs map corrections. The 3D RMS of orbit differences between dual-frequency and single-frequency orbit solutions was 9.42 cm after antenna center modeling.

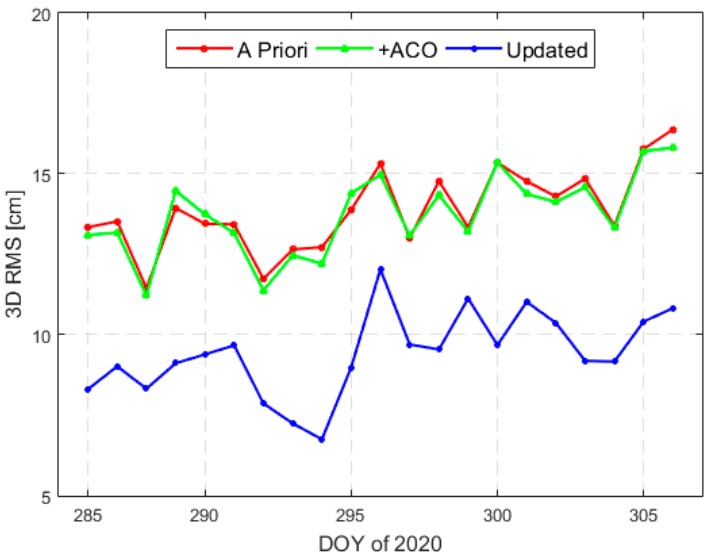

**Figure 16.** Orbit comparison between dual-frequency and single-frequency orbit solutions.

## 5. Discussion

The spherical configuration with a high area-to-mass ratio design of the Tsinghua scientific satellite is conducive to upper atmospheric density and Earth gravity field detection. The commercial off-the-shelf dual-frequency GNSS receiver is equipped as one of the main payloads for obtaining the precise orbit products. Our results show the measurements errors of the receiver at the few decimeters and few millimeters levels for code and carrier phase observations, respectively, which is competitive with other satellite missions carrying high-quality GNSS receivers. However, due to the limited system resources and microstrip patch antenna of the satellite, many long-duration GNSS gaps and large multipath errors were found. In terms of orbit determination, the orbit data with cm level precision were obtained by the receiver, which could meet the orbit precision requirement and has a better high-precision level of orbit accuracy compared with the other micro satellite missions carrying low-cost GNSS receivers. It is expected to also apply to other small-scale science missions.

In order to further improve the accuracy and reliability of orbit determination, the antenna center modeling, which is employed to estimate the offsets of ACO z-component and ACVs maps using in-flight data, is considerable for both dual-frequency and single-frequency POD. The inconsistencies at a level of 2 cm and 10 cm were found in antenna center corrections for dual-frequency IF carrier phase observations and single-frequency GRAPHIC observations, respectively. The precision of orbit determination was significant improved after using the ACOs and ACVs maps corrections. The consistencies on the 5 cm level and 10 cm level were confirmed by using ACO and ACVs map corrections for dual-frequency and single-frequency POD, respectively. Moreover, the obtained empirical accelerations in reduced-dynamic POD were reduced, and the estimated CD and CR scaling parameters were more stable and authentic. The results will have an important reference value for improving the high-precision inversion of atmospheric density and gravity field.

## 6. Conclusions

We present the performances of the GPS measurements and the results of satellite POD using in-flight data from DOY 285–305 of 2020. Regular long-duration gaps up to 50 min were observed in the GPS data, and the typical data availability was about 60–70% in each day. The average RMS values of C1 and P2 code noises were about 0.2–0.3 m, while the average RMS of carrier phase noises of L1–L2 combinations was 2.4 mm. For receiver antenna center modeling, the offsets of ACO z-component and ACVs maps for both dual-frequency and single-frequency POD were obtained using in-flight data. The average offsets of the ACO z-component were 2.04 cm and 1.71 cm for dual-frequency IF carrier phase observations and single-frequency GRAPHIC observations, respectively. The ACV maps have the characters of systematic deviations at a level of 2 cm and 10 cm, respectively.

Significant improvements were found when the orbits were determined considering the ACOs and ACVs maps corrections. For dual-frequency POD, the RMS values of post-fit residuals were reduced from 1.2 cm to 1.1 cm. After antenna center modeling, the 3D RMS of orbit overlap differences was 1.23 cm, which is reduced by 27%. Then, by comparing two series of dual-frequency orbit solutions obtained by using different types of empirical accelerations, a consistency on the 5 cm level was demonstrated for dual-frequency orbit solutions after applying the ACOs and ACVs map corrections. Meanwhile, the RMS values of estimated empirical accelerations were reduced by 56%, 26%, and 7% in the R, T, and N directions, respectively. The STD values of estimated CD and CR coefficients were both reduced. For single-frequency POD, the benefits of antenna center modeling were observed with similar performance in dual-frequency POD. Compared with dual-frequency orbit results, a better than 10 cm precision of single-frequency POD can be obtained.

**Author Contributions:** Conceptualization, K.S. and D.G.; methodology, K.S. and C.W.; Software, K.S., C.W., and K.W.; Data curation, Z.W. and Y.C.; Formal analysis, K.S., K.W. and D.P.; Validation, C.W. and D.P.; Funding acquisition, D.G. and Z.W.; Writing—original draft, K.S. and C.W.; Writing—review and editing, D.G., K.W. and D.P. All authors have read and agreed to the published version of the manuscript.

**Funding:** This work is supported by the National Natural Science Foundation of China (41874028) and the Beijing Natural Science Foundation (1224039).

**Data Availability Statement:** The data used to support the findings of this study are available from the corresponding author upon request.

**Acknowledgments:** We are very grateful to the CODE for providing the GPS precise orbit and clock products.

**Conflicts of Interest:** The authors declare no conflict of interest.

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
