# Peer review of "Tsinghua Scientific Satellite Precise Orbit Determination Using Onboard GNSS Observations with Antenna Center Modeling"

_remotesensing, doi:10.3390/rs14102479_

Round 1

Reviewer 1 Report

In this paper, the authors analyzed the qualities of GPS measurements collected from the GNSS receiver equipped on the Tsighua Scientific Satellite and showed improved POD results by considering ACOs and ACVs maps corrections. However, I think this paper provides test results only without sufficient analysis.

Page 2, lines 83-84:

  • More detail descriptions on the antenna center modeling are required if there exists any difference when compared with the methods mentioned in the previous studies.

Page 7, lines 186-187:

  • The data are collected from the dual-frequency receiver. Then, what is single-frequency solution for? Provide more specific purpose of single-frequency data processing strategy.

Page 8, lines 204-205:

  • How do you choose values for constraints for dual- and single-frequency PODs.

Page 13, lines 316-317:

  • The orbits are obtained by using different strategies but basically same GPS datasets are used. So, I think the evaluation on the precision of the solutions should be performed by using the reference truth value if available.

Author Response

Thanks for your patient review work and valuable comments on this manuscript. According to your suggestions, all corrections have been made in the revised manuscript and the followings are point-by-point responses.

Reviewer 2 Report

This is a very interesting topic and the study presents new discoveries to the community. After clarifying several misleading places, I suggest that the manuscript should be published.

The authors mentioned several times the "ground test" regarding the antenna PCO&PCV calibration, please give more information.

Please be aware that only phase center offset & variation (PCO&PCV) matters to the precise GNSS analysis, whereas the code offset & variation is much less significant due to (1) the relatively small value and (2) the relatively large observation noise. In the introduction part (starting at around line 66), clarify this. In the study of Kersten & Schon, their focus is the ambiguity resolution and the receiver & antenna with extremely large code center variations. The study of Gu et al., they only discussed PCO&PCV, nothing about code center variations.

Also, the code center offset&variation might be more important if you fix the ambiguity. The authors should clarify whether the ambiguities are fixed or not.

Line 148, "shadowing and/or reflections", from what instrument?

Figure 7b, how to explain the two blue lines in high elevation?

Line 172-176, what do you mean by L1-L2 combination? L1 minus L2? In this case, how do you consider the ionospheric delay effect?

As you estimate piecewise constant acceleration in the radial component simultaneously with the PCO-Z, would it cause correlation?

Line 243-267, please explicitly describe the APC, do you mean phase or code? If their combination, then which contributes more to the differences presented here? This part is really confusing.

Section 4.1, in your "orbit fitting", what do you actually fit? What are the observations and how do you model the fitting? This part is not clear at all.

Author Response

(The authors gave the same response as above.)

Round 2

Reviewer 1 Report

The resubmitted manuscript is properly modified based on my comments.  I think it is sufficient to meet the quality for publication.

Author Response

Thanks to the reviewer for your professional review work.